



# Evaluating the utility of qualitative data in precipitation reconstruction in the eighteenth and nineteenth centuries.

Alice Harvey-Fishenden[1], Neil Macdonald[1]

[1]Geography and Planning, University of Liverpool, Liverpool, L69 7ZT, UK

Correspondence to: Alice Harvey-Fishenden (aohf@liverpool.ac.uk)



**Abstract**. To date few studies have reconstructed weather from personal diaries. In this paper, we consider different methods of indexing daily weather information, specifically precipitation, from eighteenth and nineteenth century personal diaries. We examine whether there is a significant correlation between indexed weather information and local instrumental records for the period, thereby assessing the potential of discursive materials in reconstructing precipitation series. We demonstrate the

potential for the use of diaries that record weather incidentally rather than as the primary purpose, and the value and utility of diaries which cover short periods when used alongside nearby contemporary diaries. We show that using multiple overlapping personal diaries can help to produce a more objective record of the weather, overcoming some of the challenges of working with qualitative data. This paper demonstrates indices derived from such qualitative sources can create valuable records of precipitation. There is the potential to repeat the methodology described here using earlier material, or material from further

away from extant instrumental records, thereby addressing spatial and temporal gaps in current knowledge globally.

## 1 Introduction

There has been an increased recognition in recent years of the value of long-instrumental series spanning several centuries (Brönnimann et al., 2019; Dobrovolný et al., 2010; Todd et al., 2015), which can provide valuable information on both climate variability (Murphy et al., 2018), and sensitivities in  early records/long-term reconstructions (Murphy et al., 2020). Long

series also provide value through increased robustness in back casting in climate model testing (Talento et al., 2019), extreme event contextualisation (Todd et al., 2015; Wetter et al., 2014) and for examining social and cultural changes and modifications (Pfister et al., 2010). This has also coincided with public science projects which have seen extensive archival materials transcribed and reanalysed through both national and international programmes and initiatives (e.g. ACRE) (Allan et al., 2016; Brohan et al., 2009).

Whilst the focus has been on identifying instrumental datasets, considerable information is stored within qualitative archival source materials (Strauss and Orlove, 2003), which are more challenging to analyse using citizen science approaches. Such descriptive materials incorporate valuable information detailing not just the weather, but also human interactions with the weather and wider environment, documenting the social, cultural and economic responses to past extremes. In addition, they may also offer insights and information on more mundane intervening phases, periods during which important actions may be

taken that either exacerbate or mitigate the hazards and risks presented to communities during extremes. There have been many studies of weather diaries i.e. daily records primarily concerned with the weather (Brázdil et al., 2019a; Domínguez-Castro et al., 2015; Druckenbrod et al., 2003; Sanderson, 2018; Walsh et al., 1999), with the earliest such diary being kept in 1337-1344 (Lawrence, 1972), and weather information often recorded alongside related information, such as tides (Woodworth, 2006).



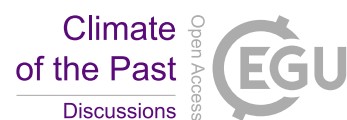

The depth and quality of descriptive materials have long been recognised within the historiographical and geographical disciplines (Oliver, 1958), but have to some extent been shunned within the sciences, though they have received increased recognition in recent years (Sangster et al., 2018). The sciences often present a preference for 'instrumental' information, because of concerns relating to quality, replicability and comparability of content within discursive materials; as (Adamson, 2015) notes, weather recording in personal diaries can lack rigour, be sporadic and be affected by the identity, personality and beliefs of the writer. However, the potential of qualitative materials is considerable, and have been shown to offer good correlations, comparable to adjacent instrumental series (Macdonald et al., 2010) and have previously been used for filling gaps in instrumental rainfall data. The potential of such sources is considerable, presenting opportunities to extend further back in time and cover areas poorly represented by instrumental information.

To date few studies have reconstructed weather from personal diaries, choosing instead simply to extract and present this weather data as it appears in the diary (Adamson, 2015; Schove and Reynolds, 1973). The series presented in this paper correspond to time periods when two diarists are writing from similar locations, and therefore offers the opportunity to test whether having multiple diarists recording simultaneously can between them produce a more objective record of the weather. Using multiple overlapping personal diaries may help to counteract the biases of personal diaries and provide a more reliable weather record. This differs from previous studies that have often focused on either a single diary (Lawrence, 1972; Sanderson, 2018) or used multiple diaries for the purpose of extending the temporal timeframe (Walsh et al., 1999). We consider different methods of indexing daily weather, examine whether there is a significant correlation between indexed weather information and local instrumental records for the period, thereby assessing the potential of discursive materials in reconstructing weather series.

## 2. Qualitative Historical Materials

Over seventy years of daily qualitative weather data between 1770 and 1865 for the UK were identified from diaries, letters and similar sources at the Staffordshire Record Office, with a wide spatial and temporal extent. The longest near continuous (daily) series in the materials collected were for Scotland 1790-1794, London 1794-1795 and 1797-1801, and Tretham in Staffordshire mid-1816-1865 (Figure 1), the majority of which are derived from personal diaries. These were not principally weather diaries, as they do not formally record daily weather, but rather a record of daily life reflecting personal recollections and thoughts of the individual from the day, which incidentally refer to the weather (particularly precipitation) daily (personal diaries with incidental weather recording).





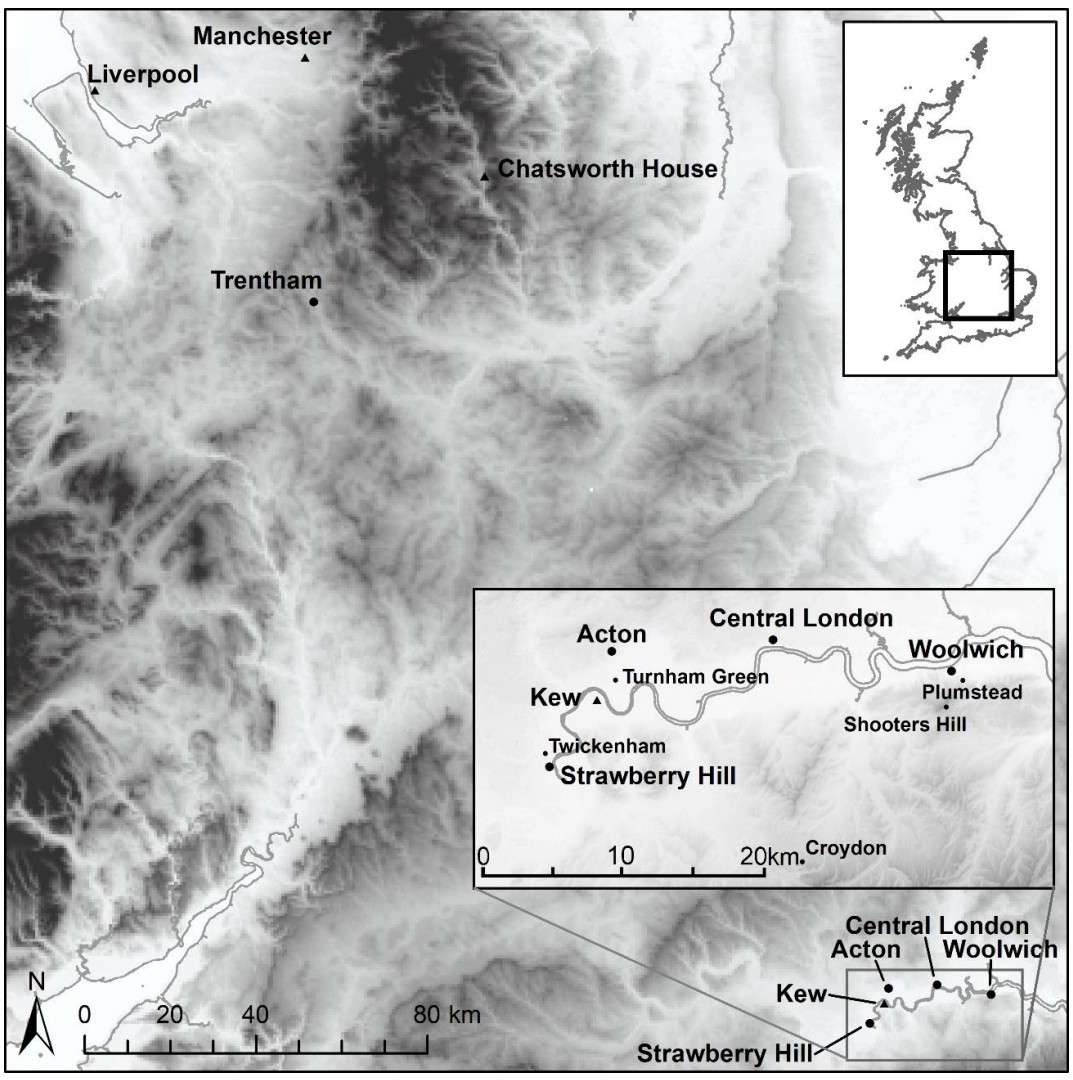

**Figure 1. Location of sites rainfall records used (background map data from Digimap, © Crown copyright and database rights 2019 Ordnance Survey (100025252))**

For the London series, diaries belonging to the author, Elizabeth Hervey (1749-1820), provided many of the daily weather
records. Hervey owned a house at Acton, in modern Greater London, and rented a house in Central London, but travelled
extensively and many of her diaries relate to her travels, in the UK and throughout Western Europe. She records the weather
daily alongside a detailed account of her day, her health and descriptions of the places she visits. A daily entry can run over
several pages and the weather is often referenced multiple times during the day, interspersed with other information. The
detailed accounts of the weather provided by Hervey are particularly notable as an early female weather recorder, with few
comparable recorders in the late eighteenth century; notable exceptions being Margaret Mackenzie's temperature series (1780-





1805) from Delvine, Scotland (Wheeler, 1994) and Constantia Orlebar's weather book (1786-1808) from Ecton, Northamptonshire (Manley, 1955). The diaries of Richard Wilkes Unett also contribute to the London series.

For the Staffordshire (Trentham) series the major sources were from the Marquis of Stafford's Trentham Estate in Staffordshire. These included the Trentham Farm reports which provide daily weather descriptions (predominantly concerning

precipitation) from 1816 onwards, with temperatures recordings from 1821, and a memoranda book belonging to the Agent for the Trentham Estate, William Lewis. The farm reports were produced monthly and had sections on the weather and reports from different employees on the estate, such as the gamekeeper. Daily entries in Lewis's memoranda book are relatively short (several to a page) but recollect the weather and its impact on Lewis's activities, and those of the wider estate. Letters between Lewis and his superior, James Loch were also consulted, as were diaries and letters from the wider North Staffordshire area

for comparison, however these offered a less complete coverage.

The Scotland series, which will not be analysed here simply because of the geographical distance from the other two series are mostly covered by a single diarist, Richard Wilkes Unett (1765-1815), but offer future interesting opportunities for further analysis alongside the records of Margret MacKenzie of Delvine, Perthshire (Wheeler, 1994). Unett travelled extensively with the military, with records according to his current posting. Although originally from Staffordshire, much of his diary material

records life in Scotland or London. He kept a daily journal recording brief notes about his daily activities, the weather, his health and the state of his garden. Interestingly, a small section of journal with weather reports kept by his father Thomas Unett also survives, but only for June of 1774 (SRO D3610/4). Unett's diary entries are generally short and factual, with information relating to the weather easy to identify and extract.

Two sections of collected data were selected for further analysis. Those from London (1797-1801) and North Staffordshire

(1816-1865) which contain overlapping accounts from different sources, and permit an assessment of how many weather records are needed per month for the greatest reliability. The shorter series for London was used to test different types of indices and the Trentham series was used to evaluate how well the most successful indices systems from the London tests could be used over a longer time period.

## 3. Indices and Instrumental Rainfall Data

No single indices system is universally used with daily weather data, though most use either a 5 or 7-point scale, such as that used by (Nash et al., 2016). Indices to date are generally not used for daily weather data, although Brázdil et al. (2019b) calculated daily days of rainfall for a set of weather diaries and converted these to a 7-degree indices using a regular distribution of ranked monthly totals. Since these indices rely on the concept of 'normal', they are difficult to apply to daily weather data, particularly when working with subjective personal accounts. With this study overlapping daily weather data from different

sources were studied, for a given year there may be >700 statements. The indices therefore need to be easy to apply, and a system which quantifies daily rainfall and/or temperature is needed, which can be averaged for months with greater/fewer





records. The system applied by Macdonald et al. (2010) uses indices from 0 (no rain, hot/drought) to 5 (very wet, storm), this approach was selected for testing (Approach A). Additionally, an indice based on days of rainfall per month used by several previous studies is also considered (Ayre et al., 2015; Brázdil et al., 2019b; Lee and MacKenzie, 2010). Two versions of this

are tested; one which assigned a value of 0 for very light rain or fog, and a value of 1 to any considerable rainfall (B) and a second which introduced nuance, aiming to capture the heaviness of the rainfall (i.e. by assigning a value of 0 to no precipitation, 0.25 to very light rain or heavy fog, 0.5 to showers or light rain and 1 to heavy rain) (C). This gave three systems of indices for consideration with the weather records from London, systems: A (after Macdonald et al., 2010); B (based on days of rainfall) and C (based on days of rainfall with consideration of heaviness of rainfall). There was insufficient data for

London (only 56 months) to test the impact of converting into a 7-degree classification following the methodology of Brázdil et al. (2019b).

It is important to verify the data, through comparison of indices and instrumental rainfall wherever possible (Brázdil et al., 2018). Fortunately, analysed and homogenised precipitation series covering the period of the diaries are available from sites with ~50km of the diary locations.

For London the nearest instrumental rainfall data covering this period comes from Kew Gardens (Todd et al., 2013). The greatest number (692) of the London records are from Elizabeth Hervey's home at Acton, (~2km (all distances from Kew)), 540 from an unspecified location in London (although in most cases this is likely to central London, (~15km), 533 from Woolwich (~30 km), 51 from Strawberry Hill (~5km), and a further 13 from various places around the Greater London area (Croydon, Plumstead, Shooters Hill, Turnham Green, Twickenham; Figure 1). The area is relatively flat and the weather

recorded in the diaries of Elizabeth Hervey at Aston and Richard Wilkes Unett at Woolwich, is generally very similar. For example, on the 10th March 1798 at Woolwich Richard Wilkes Unett wrote that it was 'A fine day. About 8 in the evening it began raining.' (SRO D3610/12/3), while at Acton Elizabeth Hervey wrote 'A beautiful morn[ing]… It rained violently this even[ing].' (SRO D6584/C/79). There are, however, occasions when Elizabeth Hervey compares the weather at her home in central London with the weather at Acton and identifies differences in rainfall, for example the 31st May 1798 when she writes

'A morn[ing] that threatens rain… Tho' much rain fell here to day and yesterday, there was scarcely any at Acton, so that my Hay has not suffered at all' (SRO D6584/C/81). As most of the non-instrumental rainfall data for London comes from very close to Kew Gardens, it is expected that there will be a high degree of correlation between the indices and the recorded rainfall.

For the Trentham series, the nearest instrumental weather station is Chatsworth House in Derbyshire (Harvey-Fishenden et al., 2019) ~50km away, however Chatsworth is located on the other side of the Peak District (Figure 1), so the Trentham data was

also compared to rainfall from Manchester (~55km) and Liverpool (~70km) (both Macdonald, unpublished). It is likely that the correlation between rainfall at Trentham and instrumental weather stations will not be as strong as the rainfall at Kew and London, simply a reflection of the distance and topography between recorder and instrumental station, though there may be seasonal variations reflecting precipitation generating mechanisms. There is a good correlation between rainfall for the period 1816-1865 at Chatsworth and Manchester (r = 0.655, p = <0.001), Chatsworth and Liverpool (r = 0.637, p = <0.001) and a





strong correlation between Manchester and Liverpool (r = 0.875, p = <0.001). The role of snowfall is challenging to quantify

and assess (Manley, 1958a), however long snowfall records are currently receiving renewed interest (Spencer et al., 2014),

particularly within early records and long series (Murphy et al., 2020), in London a long snowfall series reconstructed by

(Manley, 1958b, 1969) from 1668-1960 exists for consideration, with the series currently being updated and reanalysed.

## 4. Testing Correlation with Instrumental Rainfall Data for London

For 36 of the 61 months of the London series both Elizabeth Hervey and Richard Wilkes Unett are in London and keeping

their diaries. During the period 1796-1801 (Figure 2), four months are missing all records (July and August 1797, September

and October 1798), three months had less than 16 days weather recorded (September 1799, October 1801 and December 1801)

and a further nine months had less than 28 days recorded (December 1796, April 1797, June 1797, September 1797, October

1797, November 1798, August 1799, July 1801 and September 1801). The period analysed has a total of 2379 daily weather

descriptions, each of which was given a indices score using the three different index classification systems.

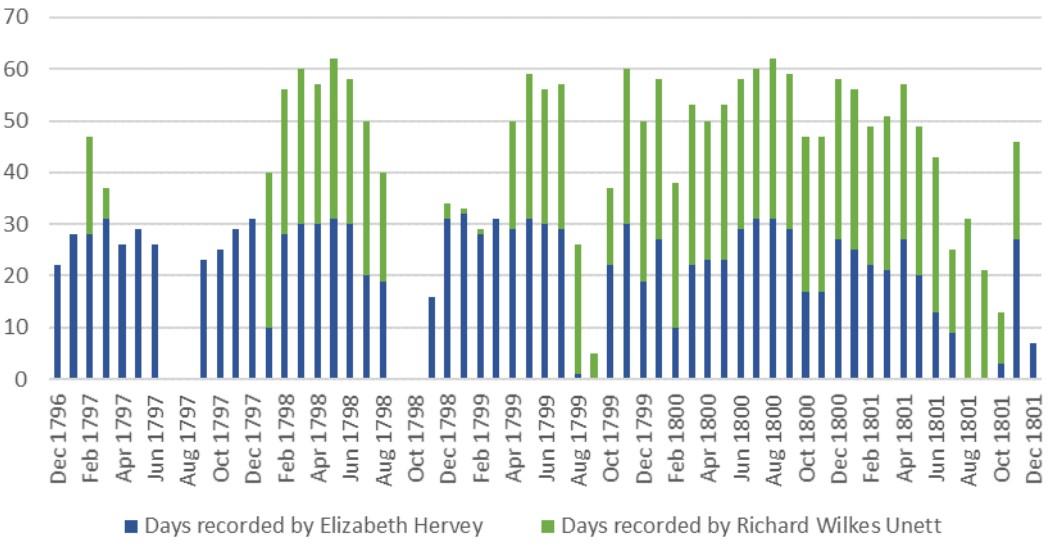

**Figure 2. Days at London (Dec 1796-1801)**

Initially any month with at least 16 days of weather data was included, the process was subsequently repeated with only months

including at least 28 days, to assess the sensitivity to record density within a month. 16 days was select as it represented >50%

of a given month and potentially accounts for an assumption (by the diarist) that the weather during the month was not

dramatically different (non-notable), thereby making this sufficient to indicate wetness/dryness of the month. 28 days was

selected as the maximum number of days that could be required, without systematically excluding February from the analysis.



As different numbers of days are recorded per month within the diaries, to aid comparison these were standardised. The sum daily index value was calculated, each day was divided by the number of documented days within the month and multiplied by the total number days in that calendar month (e.g. using index classification A for the month of December 1796, the sum 0-5 index score for each day was calculated (49, with 22 days of records that month from a possible 31, so 49 was divided by 22, and then multiplied by 31 to get a scaled monthly value of 69). The values for each of the three indices systems being assessed and each of the data levels was plotted (Figure 3a-f). All correlations were significant ($p = <0.001$) using Pearson's product-moment correlation co-efficient (r), with classification C with 28 days or more of data per month produced the best result (Figure 3f). Correlations were better than some of those between neighbouring instrumental weather stations (e.g. Manchester and Chatsworth). The strong relationships identified reflect the close proximity of Elizabeth Hervey's house to the Kew instrumental series but provide confidence in the approach by indicating that the methods of indexing the weather records produce a good estimate of the rainfall. Each of the classifications assessed produced good correlations, which were comparable or stronger than those found by similar studies e.g. Linderholm and Molin (2005) analysed the relationship between summer weather reconstructed from a Swedish diary and tree rings (R = 0.59), while Zhang et al. (2013) considered 20 years of overlapping qualitative information with instrumental data (R = 0.67). The strongest correlation identified was classification C, therefore this approach was selected for trialling on the longer dataset.







**Figure 3: London Rainfall using 3 different indexes (A-C), a,b,c,d,e and f a) 16 days index A; b) 28 days index A; c) 16 days index B, d) 28 days index B, e) 16 days index C, f) 28 days index C.**





## 4. Analysis of the Trentham Records for North Staffordshire

Compared to the London weather information analysed (Figure 3), there is more data covering a longer timeframe for Trentham in North Staffordshire. The Trentham series spans nearly fifty years, from August 1816 to December 1865. Overall, there were 20,657 weather records for Trentham for this period, with each account assigned a value from 0-1, following classification C
as tested with the London data.

The threshold for inclusion in the analysis at Trentham was set at 20 days per month, whilst two levels were assessed in London, with more days (28) presenting a stronger correlation with the instrumental precipitation series, selection of the higher threshold at Trentham would leave several gaps, including 1827-1833 (apart from 1832, William Lewis's memoranda book and letters are the only available source material), September 1818, September 1831, July 1858 and the whole year for 1862.
Therefore the threshold was lowered to 20 days, this pragmatic reduction in threshold resulted in greater temporal coverage, with the results from London suggesting the impact on correlation would be relatively limited, with only five months containing insufficient records to meet the revised lower threshold, February 1831 (which only had 18 days), August 1831 (9 days), October 1831 (11 days), October 1833 (11 days) and November 1833 (18 days) (Figure 4).

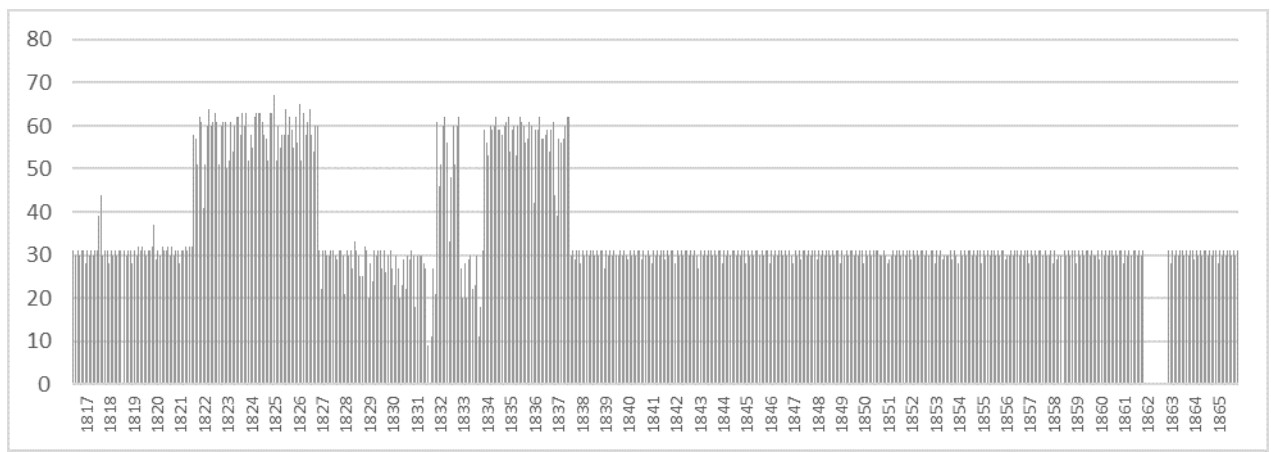

**Figure 4. Records of daily weather per month at Trentham (August 1816-1865)**

The resulting indices were compared to the instrumental precipitation series from Chatsworth, Manchester and Liverpool (CML). All three showed significant correlation (p = <0.001), with Chatsworth the weakest (r = 0.579), Manchester (r = 0.664) and Liverpool (r = 0.667) stronger (Figure 5). There is an even stronger correlation between the average of all three instrumental stations and the indices generated at Trentham (r = 0.706, p = <0.001); a potential explanation may be that
averaging has a smoothing effect on the data, reducing/removing localised extremes. While notable wet and dry periods in the instrumental precipitation series (Figure 6) correspond well with the Trentham indices, there are discrepancies in the distribution of data. There are much greater extremes of heavy rainfall seen in the instrumental precipitation from Chatsworth, Manchester and Liverpool, as extremes are hard to capture from descriptions in archive documents, because of different



people's perceptions of heavy or light rain, and the potential for observers to miss-describe the weather if they have spent key
periods of the day inside, or if the heaviest rainfall happened over night.

**Figure 5. Trentham Rainfall Indices alongside instrumental rainfall from Chatsworth, Manchester and Liverpool. 1a shows the correlation between the Chatsworth rainfall and the Trentham indices, r= 0.579 (p= <0.001); 2a between Manchester rainfall and Trentham indices, r= 0.664 (p= <0.001); 3a between Liverpool rainfall and Trentham indices: r= 0.667 (p= <0.001). 1b, 2b and 3b plot the Trentham indices against the recorded rainfall at Chatsworth, Manchester and Liverpool respectively.**

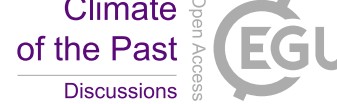

**Figure 6. Wet (blue) and dry (red) months August 1816-December 1865, with distribution (note data for 1862 at Trentham is missing)**



An assessment of the methodology applied by Brazdil et al. (2019) (using the ranking of the months by days of rainfall, and
giving them indices from -3 to 3 was undertaken, with 8.3% of the months receiving the most extreme indices, and all other
indices being 16.6% of the months). This resulted in a loss of detail and slight reduction in the correlation with the instrumental
series from Manchester (r=0.65).

## 5. Extreme Weather: Indices as Recorders of Snow, Rain and Droughts

Extremes present challenges even within instrumental series, whether it is defining the absence, excess or form of precipitation,
with considerable efforts still being made to improve and reliably identify extremes in rainfall (Archer and Fowler, 2018;
Miller et al., 2013) and snow measurement (Kay, 2016). These same challenges exist within descriptive accounts of the
weather, whilst general wet and dry phases are reliably captured, extremes, or at least the extremeness of rainfall can be difficult
to capture.

### 5.1 The problem with snow

There has long been an awareness of the challenge of under-catch in the early instrumental record. In 1891, George James
Symonds (founder or the British Rainfall Organisation) spoke about the history of rain gauges in an address to the Royal
Meteorological Society (Symons, 1891). He noted that prior to introducing Snowdon pattern rain gauges in 1864, there was a
large under-catch of snow due to absence of a protective rim on the rain gauge. The instrumental data examined here extends
to 1865, so was collected prior to the introduction of these gauges. Recently, work has challenged widely reported long term
trends in precipitation for England in Wales, by showing that much of the trend towards wetter winters and drier summer can
be explained by under catch in the early record (Murphy et al., 2020). It is important, therefore, to consider the contribution of
snow to this data. An assessment of snow days and months with snowfall at Trentham was undertaken, as evidenced in the
diaries, between October and May. Most years had at least some snow, with the greatest snowfall in January, with an average
of 1.9 days, closely followed by March with 1.8 days, and then February with 1.6 days, all other months have less than a day
of snowfall on average; December (0.9), April (0.8), November (0.5), May (0.2) and October (0.1). Most snow in the records
is in winter (December-February), with an average of 4.43 days per winter, followed by the Spring (March-May; 2.76 days)
and autumn (September-November 0.66 days). Visual examination of the Trentham indices against recorded rainfall at
Chatsworth, Liverpool and Manchester showed that months with snowfall tended to have less rainfall in the instrumental
record than was suggested by the indices.

Separating out the months with and without snow gives a stronger correlation to the Manchester precipitation series for the
months without snow (r= 0.71), and a similar correlation for months with snow (r=0.63) (Figure 7). The indices are
systematically indicating slightly more precipitation than is recorded by the rain gauges. There are four possible reasons for
this systematic difference between the records:





i.          The potential of under-catch in the instrumental record (e.g. (Murphy et al., 2020).

ii.          It might reflect over estimation of precipitation due to overreporting of snow. Snow is a highly visible weather phenomenon, and therefore might be over reported (Spencer et al., 2014). There is also the complication of whether, when a record says 'snow', whether it is referring to new snowfall, or snow lying on the ground.

iii.          It may reflect over estimation of precipitation resulting from the way snow has been translated into indices. Snow has been treated in the same way as rain in the indices system, but it is possible that it should have been treated more like showers

than substantial rainfall.

iv.          The potential for greater variability in snowfall between the two sites compared to rainfall, reflecting local topographic and climatic conditions, with a slightly greater capacity for snowfall at Trentham (Barrow and Hulme, 2014; Mayes, 2000).

It is likely that a combination of these factors may account for the differences in the two series.  The indices for the 138 months with two or less days of snow had a better correlation with the rainfall at Manchester (r= 0.70) compared to the 58 months

with more than two days of snow (r= 0.51). When plotted (Figure 7), many of the outliers come from months with the medium amounts of snowfall (i.e. 2.5-4 days of snow), suggesting that overreporting of snow (perhaps the reporting of snow remaining on the ground, rather than fresh snow) may be the cause of the lower correlations for the months with more snow.

Snow can have severe impacts on farming in the UK (Jones et al., 2012). An example of this can be found in this material from Trentham in the winter of 1819-20; with snow lying on the ground between the 30 December 1819 and the 29th January 1820,

with a thaw beginning on the 22nd January causing flooding (SRO D593/K/3/2/2, SRO D593/L/6/2/2 and SRO D593/K/3/2/1). On the 3rd January 1820, William Lewis recorded that the average depth of the snow was 22 inches (0.56m) (D593/L/6/2/2). Lewis's letters highlight some of the impacts of this extended period of cold weather. On the 17th January he wrote:

*"The Storm continues and no appearance of any alteration. The Snow has been ever on the ground which causes both Sheep and Cattle to be fed out of doors with every morsel they consume"* (letter sent by William Lewis to James Loch (SRO

*D593/K/3/2/2).*

While towards the end of the episode he records that:

*"[T]he severe weather has completely put all out door work at a stand for some weeks which is the cause of the present distress [amongst the parishioners] a moderate thaw has now taken place which I trust will continue"* (letter from William Lewis to James Loch, 23rd Jan 1820 SRO D593/K/3/2/2).*

The long lying snow led to underemployment (Lewis notes that he will take on more labourers for ditching and draining when the weather allows), and higher costs in cattle farming. While creating indies allows weather to be contextualised and compared with instrumental records the original qualitative weather records tell of impacts and add details that are absent from the indices alone.




**Figure 7. a) Days of snow per month (note data for 1862 is missing) and indices for months with (b), and months without (c) snow, plotted against Manchester rainfall. For the months with snow r=0.63, for months without snow r=0.71 (for all correlations p=<0.001).**



## 5.2 Heavy rainfall

While indices are good at capturing dry weather and moderate rainfall events, they appear to be weaker for capturing heavy rainfall. This is an inherent problem of descriptive records, which often lack clear distinctions between moderate to heavy and then extreme rainfall. There is a large range in amounts of rainfall which could fall during 'rain' or during a 'shower'. When a lot of rain falls in a short period of time, indices will generally underestimate precipitation, since the maximum value that can be given for a day is 1. For this reason, studies that have collected similar data have thought it inadvisable to attempt conversion into quantitative values (Lee and MacKenzie, 2010). The difficulties in accurately estimate extremes and the problems with trying to convert qualitative information into quantitative values is demonstrated by comparison of the average of the three stations (Chatsworth, Liverpool and Manchester) and the Trentham indices (Figure 8), to produce a conversion from indices to rainfall (millimetres). Compared to the instrumental data from CML stations the minimum value of the converted indices is higher (14.48mm, compared to between 0mm and 3.54mm) and the maximum value much lower (123.78mm, compared to between 190.95-282.35mm) and the standard deviation lower (21.26mm, compared to between 31.43-35.05mm).



**Figure 8. Rainfall (in mm) at Chatsworth, Manchester and Liverpool, and an estimate of the rainfall (in mm) at Trentham based on the indices**

Unlike the issues around snow where there are possibly multiple contributing factors resulting in differences between the indices and recorded instrumental rainfall, it is apparent here that the main issue with extremes of rainfall is around the lack of range in the indices, particularly when it comes to heavy rainfall. It might, however, be possible to address the deficiencies of the indices at representing extreme rainfall by adjusting the indices. For example, they might be improved by increasing the values for months with recorded floods, or other severe impacts. This would be similar to approaches undertaken by Brázdil et al. (2019) in the application of indices, where after applying indices based on ranking of days of rain per month, they were adjusted to account for months with particularly heavy or light rainfall.

However, taking for example July 1828, this method might not always improve results. In July 1828, 282.35mm rainfall fell at Chatsworth, 251.42mm at Manchester and 190.95mm at Liverpool, whereas Trentham only experience 11.75 days of rainfall (which using a simple regression to convert to rainfall equates to 94.07mm). It seems likely, therefore, that this is a substantial




underestimation of the rainfall caused by above average rainfall on multiple days. There are six 'very wet' days mentioned in the diary descriptions, but no severe impacts of wet weather, which might flag a need for this month to be adjusted. The only impact mentioned is a delay in 'getting in the hay' (SRO D593/L/2/2b), which could be caused by even mild wet weather. For

the current data, it is hard to see how any methodology based on impacts would flag this month as one meriting adjustment.

From the perspective of quantifying and comparing rainfall, therefore, conversion to millimetres does not create a useful dataset and is also unnecessary. To better quantify and compare rainfall across instrumental and non-instrumental records, it would be interesting to compare the number of days of rain per month within the instrumental record with the days in these diaries, and to investigate the amounts of rain per day in the instrumental record. Unfortunately, most of the surviving

instrumental records for this period only contain monthly totals, not daily totals or records of the number of days of precipitation.

### 5.3 Droughts

There are several droughts which have been identified elsewhere which occurred during the period covered by the records from London and Trentham. First, there is a drought period identified by Todd et al. (2013) between 1801 and 1808; the most

severe drought episode begins in September 1802 at Kew, however it is preceded by conditions fluctuating between normal and rainfall deficit. This period of dry weather preceding drought onset is evident in the records from London. On the 20th July 1800 Elizabeth Hervey reports that 'ground is sadly parched' (SRO D6584/C/93) and on the 25th July 1800, she writes that there is 'Still burning weather the leaves fall as in winter' (SRO D6584/C/93). The leaves falling from the trees is corroborated by Richard Wilkes Unett, who on the 24th September 1800 wrote 'Owing to the very dry weather in July, most

kind of trees lost their leaves the same as in October' (D3610/12/3 p139). Notably low rainfall in the Kew series of 7.9mm in February 1800 and 0mm in July 1800 is reflected in the indices for London, although the low value in February 1797 of 5.6mm is not captured as well within the indices.

In terms of the Trentham record, droughts are recorded at Chatsworth in 1821, 1826-8, 1835-6, 1844-5 and 1847-8, all of which are relatively minor droughts, within the context of the long drought series available (1760-2015; Harvey-Fishenden et

al., 2019). The farm and wood-ranger reports, which accompany the weather information recorded at Trentham , mention drought in Spring 1817, writing that the 'trees planted this spring are suffering much from the extreme drought, except the Mountain Ash planted in Trentham coppice which having a cool soil, and being shaded by the larger oaks are looking very well' (D593/L/6/2/2). The CML series all show low rainfall in January 1817, March 1817 and April 1817, while the Trentham indices show dry weather in January and April 1817 and a wet March. This reflects the challenge of historical descriptive

accounts truly capturing a 'drought' accurately following precipitation, as a short phase of rainfall may offer some respite but may not formally terminate a drought event as defined and classified using drought indices. The dry weather may no longer have an agricultural or water resource impact but may still technically be a meteorological drought.



Neither the 1821 drought nor the 1826-8 drought are reported as severe weather events in the archival records, with no drought impacts are reported at Trentham. The dry weather in 1821 seems to have led to abundant crops and fine weather, whereas in

1826-8, diary entries focused on the day to day running of the farm and there is no entry assessing the wider impact of the weather. There are two other droughts which occur during this record; 1835-1836 at Chatsworth, lasting 16 months, with a peak severity of -1.0 (using the Standardised Precipitation Index) in August 1835, and 1844-1845 at Chatsworth, lasting 15 months with a peak severity of -1.5 in March 1845. Once again, although dry, there is no comment on the weather or its impact in these records from Trentham.

**5.4 Other extreme weather**

One of the strengths of multi-source qualitative recording is that sometimes the overlap between different archival sources can tell you more about the impacts of extreme weather than any one document alone. For example, a letter sent 26th June 1824, from William Lewis to James Loch about the impacts of a storm, which reads

*"In my last I forgot to name to you that we had a severe Thunder storm on Monday last and the lightening (sic) killed six Deer*

*under a Tree close by the reservoirs & the Day following a cow at Corn Croft be-longing to one of Mr Lord's labourers"*
*(D593K/3/2/6).*

Accordingly, in the monthly report for June 1824 at the Trentham estate, we see an entry for the previous Monday, the 21st June reads 'Thunder and Rain'. In the accompanying Park Keepers report for the same month, it notes that 1 buck and 5 does were killed by lightning (although does not give a date) (SRO D593/L/6/2/2). Examination of William Lewis's own

memoranda book for the same date, however, details that he merely notes 'a very dull morning' (SRO D593/L/2/2b). The different sources tell complementary stories; the letter identifies which event killed the deer, but not the date, the monthly report gives the date for the storms, but does not state which storm killed the deer. William Lewis's memoranda book does not even mention the event at all. The notes on the weather kept by William Lewis appear to have generally been made quite early in the morning, and he appears to sometimes revisit the previous day to update it with later changes in the weather, but

sometimes, as in this case, fails to do so.

Descriptive accounts can also provide valuable insights relating in to weather events that are poorly recorded in early-instrumental records, such as tornado's, mists, fog, haars and lightning storms, with documentary sources offering a valuable tool in creating long reconstructions (e.g. Camuffo et al., 2000).

**6. 'The Year without a Summer' (1816) in Staffordshire**

In April 1815, Mount Tambora in Indonesia erupted, with impacts around the world (Pfister and White, 2018a), 1816 has been described as the 'year without a summer' by several authors (Stothers, 1984; Veale and Endfield, 2016; White et al., 2018), with the impacts of the Tambora eruption on weather extensively recorded across Europe. The diaries of Elizabeth Hervey





offer considerable potential for examining its impact from a different perspective, that of a female traveller, as she travels around Europe during the summer 1816.

It is unfortunate that the Trentham record begin in August 1816, since it misses much of this 'summer'. June saw some of the worst impacts across Europe, with anomalously cold and wet weather (Luterbacher and Pfister, 2015). There have been several attempts to use diaries from this period to examine the effects of the Tambora eruption in the UK, such as Lee and MacKenzie (2010) who used a farmer's diary from near Manchester (~5km away), which is about 50km north of Trentham, with  wind direction, barometric pressure and observations of weather and other phenomena recorded (including red skies). The Trentham

records primarily record rainfall, and there are no descriptions of non-weather phenomena, such as red skies. Veale and Endfield (2016) used diary sources to situate the year 1816 within a UK context. They describe the general pattern of the weather nationally around this time; In August much hay was spoilt by rain, September was cold and frosty, there were floods in October, while November was wet and very cold (Veale and Endfield 2016). These conditions led to food shortages; with 1817 described as having the fourth successive cold and sunless spring, and also being dry. There was a heatwave in June,

with July, August and September wet. The records from Trentham support some of this; August to December 1816 are all relatively wet months at Trentham. This is at odds, however, with the instrumental records from Manchester and Liverpool, with August quite dry (although it is wet at Chatsworth), and September and November are either dry or average at instrumental stations. In October (1816) the Trentham farm reports state that 'grain of all descriptions much injured by the inclemency of the weather', however other crops such as beans and turnips seem to have given normal yields (D593/L/6/2/2).

The accounts identified by Veale and Endfield (2016) for 1817 are replicated in the Trentham records. March at Trentham has five full days of snow, or 'snow, hail and rain', and a further evening of rain and snow, although it appears to have only been a few inches deep. There were also fine days, some 'gentle showers' and one day which is described as 'fair- very mild'. April has ten days which are 'Fair- cold', three which are 'fair- very cold' and one which was 'very cold with a little snow'. There also fair days and drizzly days. In May there are three references to it being cold and four occurrences of hail (SRO

D593/L/6/2/2). The spring months are therefore generally, although not universally consistent with the overall pattern described by Veale and Endfield (2016). The heatwave as reported lasted from the 19th-25th June; at Trentham the 19th to the 26th June are described as 'very hot', with several thunderstorms. Nationally, July, August and September were generally wet (Veale and Endfield, 2016), however, Trentham, Chatsworth, Manchester and Liverpool follow the same pattern, with July and August wet, but September dry. Generally, the weather records from Trentham seem to fit well with the patterns described

by Veale and Endfield for the years following the Tambora eruption, although these years are not statistically different from preceding years.



## 7. Confidence in Documentary Source Reconstructions

The quantity and quality of data impact on indices. A particular issue with diaries is that, when incomplete, they tend to only report more extreme events and are biased towards snow and rainfall. For example, Table 1 shows the entries for the first half

of December 1807 from the diary of Thomas Birds (in Eyam in Derbyshire, SRO D1229/4/6/7) and James Caldwell (~45 km away at Talke, in North Staffordshire). These demonstrate that James Caldwell is only recording particularly extreme days, and does not report on 'fine' days, and if we were to produce indices based on both these sources, this month might look colder and snowier than it was. While the London data demonstrates the benefits of having diarists recording simultaneously when producing indices, the source material must record both extreme and normal days. Much of the collected data from Trentham,

from William Lewis's memoranda and the farm reports, does not include impacts, which limits its utility when considering extreme weather, however they do record day-to-day activities such as harvest or ploughing, which may be delayed due to inclement or inappropriate weather.

**Table 1.  Diary entries for Thomas Birds and James Caldwell December 1807**

| Date | Thomas Birds (Eyam) | James Caldwell (Talke) |
| --- | --- | --- |
| 1 Dec 1807 | A fine winters day | |
| 2 Dec 1807 | A plashy day & some rain | |
| 3 Dec 1807 | A very fine frosty morn | |
| 4 Dec 1807 | A most tempestuous wet day | |
| 5 Dec 1807 | Showry day | |
| 6 Dec 1807 | Snowy day | Snow. |
| 7 Dec 1807 | A severe frost | Severe frost. |
| 8 Dec 1807 | A most tempestuous snowy day | |
| 9 Dec 1807 | A fine day | |
| 10 Dec 1807 | A fine day | |
| 11 Dec 1807 | A partial thaw at home | At night great fog & Snow. |
| 12 Dec 1807 | A fine day continued thawing a little | |
| 13 Dec 1807 | A fine day | |
| 14 Dec 1807 | A very fine day | |

While there is insufficient information in the documentary records to produce temperature indices, the Trentham farm reports include temperature measurements, taken at 8am and 8pm from 1821 onwards, while both Elizabeth Hervey and Richard Wilkes Unett had access to a thermometer and occasionally reported temperatures, particularly extremes. There is one day (14th July 1800) in the London series where they both report the temperature. Richard Wilkes Unett writes 'Very close & warm. The glass today in a room where there are three doors was at 75⁰ most of the day' from Woolwich (D3610/12/3 p127),

while from Acton Lodge, Elizabeth Hervey writes 'Hottest day we have yet had thermometer 75 in the shade.' (D6584/C/93).



Unfortunately, neither of these diarists consistently record the temperature, and because they both lived relatively transient lifestyles, the readings they give are not always from the same thermometer.

In undertaking the classification, the indices were applied by one person, with 27,794 records (20,657 for Trentham, and 2,379 using three different methods for London). Previous studies have had indices applied by two or more researchers and then
been compared (Nash et al., 2016). It would be impractical to have two researchers look at this volume of material reviewed in this study. It is highly likely therefore that there are occasional errors in this; between transcription errors from volunteers who collected and transcribed some of the data, possible inconsistencies in the application of the indices (since they were not all done in one sitting) and typographical errors, there are multiple opportunities for errors to arise. However, the methods used limit the impact of such errors occurring. This kind of reconstruction of precipitation series using indices is most valuable
when no nearby instrumental records are available. This work has shown that if carefully applied these indices reconstruction approaches can have good correlation with instrumental records of rainfall and could therefore be used where no instrumental records exist, to help understand and contextualise reported impacts of extreme weather.

Different types of sources record weather differently, influencing how the resulting data can be used. Diaries often use weather as a starting point for an account of the day, as do some letters: mundane weather acts as a space filler, a neutral topic. If a
diary records the weather every day, or a letter is not primarily about the weather, a full range of different types of weather are likely to be recorded. If, however, it is a diary that only records the weather occasionally it is likely to record more unusual weather, for example heavy rain, snow or frost (Table 1). Likewise, letters where the weather is one of the main topics, are likely to record extreme weather events. The other type of weather that appears in letters, is weather used to explain actions, or inaction, in estate correspondence. For example, William Lewis wrote to his superior, James Loch in 1824; 'I followed up
the old proverb "make hay while the sun shines" the weather still remains favourable and dry and all have been busy with the Turnips which are in a very forward state and very promising.'

A further consideration is the purpose of the original diaries; most of the records are not intended to record the weather accurately for posterity but are instead a record of someone's life and thoughts. A possible criticism of weather reconstructions from diaries is that they only record the weather during the daytime (Adamson, 2015). Whilst true, all the diaries used in this
project tend to comment on heavy rain or snow if it occurs overnight, so if the object is to record rainfall, this may not always a problem.

Careful analysis of the Elizabeth Hervey diaries presents a contrasting perspective to comments by Adamson (2015) that diaries as a source of information in climate research provide 'highly personal documents…representing an unbiased account'. There is evidence in the diaries of Elizabeth Hervey of her reading aloud from her diaries for her friends and acquaintances, and of
the diaries being edited after her death by her son (with sections he considered uninteresting, or unsuitable being redacted or removed).

## 8. Conclusions

Work on diaries to date has often focussed on specific weather diaries continuously recorded for long periods (Pfister and White, 2018b), this paper demonstrates the value and utility of diaries reflecting shorter periods, particularly when used

alongside nearby contemporary diaries. It also demonstrates the great potential in the use of personal diaries which record weather incidentally rather than as the primary purpose for reconstructing long weather series. Such diaries could provide greater spatial and temporal coverage than instrumental records currently offer, presenting opportunities for extension of existing knowledge to areas where no, or limited, instrumental information exists.

The records described here could be complemented by use alongside other weather records, for example the thrice daily

weather observations made by staff at Boulton and Watt Soho Manufactory in Birmingham between 1793 and 1830 identified by Veale and Endfield (2016), or as noted previously in Scotland in comparison to Margaret Mackenzie's temperature series.

The indices approaches described are valuable in considering the influence of different weather systems and in identifying wetter and drier periods, qualitative data can provide additional information beyond instrumental records (e.g. nature of precipitation) and the subsequent responses undertaken by individuals and communities to events. This paper demonstrates

that for periods with overlap between documentary sources, indices can create valuable and reliable records of precipitation.

There is value in qualitative daily data used in the creation of the indices, since it would also allow creation of targeted indices, such as consecutive dry days, which may be more useful than monthly data for investigating specific impacts such as drought (Pfister et al., 2020). A further strength of this type of data over contemporaneous instrumental data is that it reflects a finer scale, as such it might be possible to use such information to scale monthly or annual rainfall totals from instrumental records

to the (sub-)daily scale.

**Data Availability**

The original archival materials used in this study, as detailed in the list of archive material, are available at the Staffordshire Record Office. The data used to create graphs is available from the corresponding author.

**Author Contributions**

AHF prepared the manuscript, extracted data from the archives and ran a volunteer project extracting data from the archives and produced the indices. NM provided instrumental precipitation data and contributed to the manuscript.

**Competing interests**

The authors declare that they have no conflict of interest.





**Acknowledgements**

Some of the data used in this paper was collected by a group of volunteers at the Staffordshire Record Office as part of a Collaborative Doctoral Award Studentship. The authors would like to thank the project volunteers and the staff at the Staffordshire Record Office for their support and assistance.

**Funding**

This work was supported by the Arts and Humanities Research Council under Grant AH/N005147/1

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
