# Peer review of "Evaluating the utility of qualitative personal diaries in precipitation reconstruction in the eighteenth and nineteenth centuries."

_Climate of the Past, 2020_

## Referee Comment (RC1) · David Nash (Referee) · 10 Aug 2020

Thank you for the opportunity to review this manuscript, which presents a useful methodological consideration of the use of qualitative data from personal diaries in weather reconstruction. The authors have analysed a huge amount of qualitative data – for which they should be commended – and have analysed it rigorously. The study is novel and the methods (mostly) clearly outlined. The material covered is relevant to Climate of the Past. In short, there is certainly scope for the work to be published.

While the science in the paper is sound, the presentation and written style detracts significantly from the quality of the data presented. There are many long and overly

complex sentences, the writing is (in places) imprecise, and many figure captions are vague. The results are really strong but the conclusion, if anything, downplays them.

I suggest that significant revision is needed before the manuscript could be considered suitable for an international readership. At a bare minimum, the authors should read over the full text carefully and tighten up the wording. They should split their longer sentences – e.g. lines 72-77, 133-135, 135-138, 140-143, 168-171, 181-184, 197-200, 216-218, 281-284 (there are many others) – into multiple parts. This will greatly improve the readability of the manuscript. The authors also need to check for consistency in the use of 'which' and 'that' throughout the text – often this is incorrect. Please see also my comments about the misuse of the singular/plural of index in the results sections.

Inconsistency in citations – there are places where the referencing software used in the manuscript goes awry and places parentheses around citations when the author name should be included in the text of the sentence.

Specific comments:

The title of the paper would be more accurate and informative if it included the words 'from private diaries'.

Section 2 – I appreciate that sources are cited throughout the results section, but it would be useful to refer the reader (perhaps at the end of the first paragraph) to the list of Archival Sources.

Section 4 onwards – check the use of the singular (index) and plural (indices) here, as the results sections are full of inconsistencies. "No single index system is universally used. . ." ". . .converted these to 7-degree indices. . ." "Additionally, an indexing approach based on. . ." ". . .each of which was given an index score. . ."

Figure 2 – caption could be more informative.

Line 158-162 – this is a very long and confusing explanation of the standardisation

process – consider fragmenting the sentence.

Figure 3 – the caption is very densely written and could be more informative.

Line 176 – this is the second Section 4 in the manuscript.

Figure 5 – I'm assuming that the blue line in 1b, 2b and 3b is the Trentham index series, as the caption doesn't explain this. Also, I may have missed it, but have you explained somewhere before the first mention of this figure how the index values for Trentham have been converted into mm rainfall?

Figure 6 – the distribution data at the bottom of the figure require appropriate axes and a little more explanation in the caption.

Line 247 – do you really mean 'greater capacity for snowfall at Trentham'? 'A higher propensity for snowfall' might be better.

Line 250 – is 'medium' the best word here?

Figure 7 – I assume this is Trentham as the caption doesn't state this. And indices for what (in both the caption and axes labels)?

Figure 8 – again, how are the Trentham values converted into mm rainfall?

Line 303 – one example of the misuse of 'which' here – what you mean is '. . .(drop the comma) that might flag. . .'. For comparison, the use of 'which' in the next sentence is correct.

Lines 307-311 – I was going to ask exactly this, as a direct comparison of 'missing days' in diaries (and potentially vice versa) would be very instructive.

Line 357 – spelling 'tornadoes'.

Line 365-369 – confusing sentence.

Line 375 – unclear what is meant by 'this' here.

Line 381 – unclear what is meant by 'it' here.

Lines 393-394 – I understand what you are trying to say, but this is a very UK-centric take on diaries. I could point you to numerous personal diaries from beyond the UK that are biased towards drought.

Lines 413-418 – I understand the reason for including this paragraph, but it has the potential to seriously detract from the results of the study. Much better would be to state earlier (in the methodology): (i) that one person completed the analysis for all index series to improve consistency; (ii) that volunteers were involved in transcription but there was quality control.

Section 8 – I want to end on a positive. The results of this study represent a huge amount of work and are potentially very interesting. However, the Conclusion seriously underplays the quality of the research and should be much stronger. This is an opportunity to point out the key findings of the paper and highlight which aspects of the methodology were most successful. To my knowledge, no one has conducted this volume of analyses of diary entries, so this is the opportunity to recommend a methodology to be used in future studies. This will elevate the manuscript from 'an interesting study' to a 'must-cite paper' for future researchers seeking to use diaries for climate reconstruction.

---

## Referee Comment (RC2) · Anonymous Referee #2 · 13 Aug 2020

[Evaluation] This manuscript presents very interesting research on reconstruction of rainfall amount from several simultaneous diaries in the past. The methods used are appropriate and the conclusions derived from these and the interpretations are consistent and sound. I believe the paper will be of interest to the readership of this journal and would recommend it for acceptance after the minor revisions. I look forward to seeing it in print.

[Comment] I understand that this paper deals with past climate in UK, Europe. However, in Chapter 1, I believe that related past studies should be introduced not only European cases but also other parts of the world (I could find only one reference in

[Figure]

China). As far as I know, rain day index could show more strong correlation rather with temperature than rainfall amount in several Asian countries. I would like to recommend that the authors take more information from related studies outside of Europe and reflect it in the context.

[Minor comment] Page 6, line 107-116: Please state more detailed classification and weather descriptions especially for Approach A and C. At this moment readers need to get and read references to understand the methods in detail.

Figure 1: In Figure caption, please explain what circles and triangles in the map refer.

Figure 2: In Figure caption, please provide detail explanations.

---

## Author Comment (AC1) · 21 Aug 2020

We would like the thank the referee (David Nash) for a very thorough and thoughtful review of our paper, and for his constructive comments. We are happy that the referee recognises the value of the data analysed, the quality of the analysis, the strength of the results and the novelty of the study. We are happy to act on his suggested edits and additions.

In our revised manuscript we will pay close attention to the referee's concerns regarding sentence length, citations and grammatical inconsistencies.

We note that the referee suggests the inclusion of the words ‘from private diaries’ in the title and will look to revise the title accordingly.

We will review all our captions for clarity and completeness.

We note some confusion around the conversion of the Trentham index series into mm of rainfall. We will clarify this in the appropriate figure captions (figure 5 shows the raw index series plotted on a secondary axis and figure 8 shows the conversion to mm) and in the relevant section of the text (section 6.2). We are happy to revise the labelling of axes in figures 6 and 7 as suggested.

We acknowledge that lines 392-394 presents a UK centric perspective and will modified the text to reflect this.

We will follow the referee’s suggestion, and replace lines 413-418 with a short section in the methodology discussing these issues.

We will work to improve the conclusion in the revised manuscript so that it does justice to the strength of the results.

---

## Author Comment (AC2) · 21 Aug 2020

We would like the thank the referee for reviewing the paper, and for their useful and constructive comments. We are glad that they consider the content of the paper to be sound. We are happy to incorporate their suggestions into the revised manuscript and will accordingly consider relevant non-European literature. We will add more detail where suggested in the methodology and will revise our figure captions.

---

## Author Response (AR1)

**Response to editor/reviewer comments and changes to manuscript:**

| Editor comment | Author response |
|---|---|
| The title should be changed to specify the source type (diaries) | We have modified the title accordingly. |
| The first reviewer requested stylistic edits throughout the text and clarification of the figures. | We have revised the manuscript, making use of Referee 1's detailed comments. |
| Both reviewers indicated that the manuscript should include more international context regarding weather diaries and methods for using them. | We have included more references to international usage of diaries for weather reconstruction throughout the manuscript. |
| Both reviewers requested that the conclusion be rewritten to emphasize the study's global methodological significance for historical climatology and not only its findings regarding past British climate. The last two points are particularly important in light of the international and comparative focus of this special issue. | We have extensively rewritten the conclusion. |
| **Referee 1 comment** | **Author response** |
| Thank you for the opportunity to review this manuscript, which presents a useful methodological consideration of the use of qualitative data from personal diaries in weather reconstruction. The authors have analysed a huge amount of qualitative data – for which they should be commended – and have analysed it rigorously. The study is novel and the methods (mostly) clearly outlined. The material covered is relevant to Climate of the Past. In short, there is certainly scope for the work to be published. While the science in the paper is sound, the presentation and written style detracts significantly from the quality of the data presented. There are many long and overly complex sentences, the writing is (in places) imprecise, and many figure captions are vague. The results are really strong but the conclusion, if anything, downplays them. | Response to Referee 1 (David Nash) We would like the thank the referee (David Nash) for a very thorough and thoughtful review of our paper, and for his constructive comments. We are happy that the referee recognises the value of the data analysed, the quality of the analysis, the strength of the results and the novelty of the study. We are happy to act on his suggested edits and additions. |
| I suggest that significant revision is needed before the manuscript could be considered suitable for an international readership. At a bare minimum, the authors should read over the full text carefully and tighten up the wording. They should split their longer sentences – e.g. lines 72-77, 133-135, 135-138, 140-143, 168-171, 181-184, 197- | In our revised manuscript we have paid close attention to the referee's concerns regarding sentence length, citations and grammatical inconsistencies. We have assessed every occurrence of the words 'that' and 'which' and hope that they are now used correctly. We have done the same for 'index' and 'indices'. We have checked all the citations. |

| | |
|---|---|
| 200, 216-218, 281-284 (there are many others) – into multiple parts. This will greatly improve the readability of the manuscript. The authors also need to check for consistency in the use of 'which' and 'that' throughout the text – often this is incorrect. Section 4 onwards – check the use of the singular (index) and plural (indices) here, as the results sections are full of inconsistencies. "No single index system is universally used…" "…converted these to 7-degree indices…" "Additionally, an indexing approach based on…" "…each of which was given an index score…" Inconsistency in citations – there are places where the referencing software used in the manuscript goes awry and places parentheses around citations when the author name should be included in the text of the sentence. | |
| The title of the paper would be more accurate and informative if it included the words 'from private diaries'. | Accept – we have revised the title to reflect the sources and added a sentence to the Introduction (para 3) differentiating between private and personal diaries. |
| Section 2 – I appreciate that sources are cited throughout the results section, but it would be useful to refer the reader (perhaps at the end of the first paragraph) to the list of Archival Sources. | Accept - We have added the sentence 'Details of archival sources used in this reconstruction can be found at the beginning of the reference list' to the end of section 2, paragraph 1. |
| Figure 2 – caption could be more informative. | Accept - We have reviewed all our captions |
| Line 158-162 – this is a very long and confusing explanation of the standardisation process – consider fragmenting the sentence. | Accept - We have revised this sentence, and hope it is now clearer. |
| Figure 3 – the caption is very densely written and could be more informative. | Accept – we have rewritten this caption. |
| Line 176 – this is the second Section 4 in the manuscript. | Accept – corrected. |
| Figure 5 – I'm assuming that the blue line in 1b, 2b and 3b is the Trentham index series, as the caption doesn't explain this. Also, I may have missed it, but have you explained somewhere before the first mention of this figure how the index values for Trentham have been converted into mm rainfall? | Accept - We note some confusion around the conversion of the Trentham index series into mm of rainfall. We will clarify this in the appropriate figure captions (Figure 5 shows the raw index series plotted on a secondary axis, and Figure 8 shows the conversion to mm) and in the relevant section of the text (section 6.2). |
| Figure 6 – the distribution data at the bottom of the figure require appropriate axes and a little more explanation in the caption. | Accept - We have revised the caption and labelling of axes for figure 6 as suggested. |

| | |
|---|---|
| Line 247 – do you really mean 'greater capacity for snowfall at Trentham'? 'A higher propensity for snowfall' might be better. | Accept - We have changed this sentence as suggested by the referee. |
| Line 250 – is 'medium' the best word here? | Accept |
| Figure 7 – I assume this is Trentham as the caption doesn't state this. And indices for what (in both the caption and axes labels)? | Accept - We have revised the axes labels and caption. |
| Figure 8 – again, how are the Trentham values converted into mm rainfall? | See section 6.2 paragraph 1 |
| Line 303 – one example of the misuse of 'which' here – what you mean is '…(drop the comma) that might flag…'. For comparison, the use of 'which' in the next sentence is correct. | Accept - This sentence has been modified. |
| Lines 307-311 – I was going to ask exactly this, as a direct comparison of 'missing days' in diaries (and potentially vice versa) would be very instructive. | Agree – we have added a further comment to this section on probabilistic fittings, but recognise the limitation here is the data resolution during early instrumental records. |
| Line 357 – spelling 'tornadoes'. | Accept - This typographic error has been corrected. |
| Line 365-369 – confusing sentence. Line 375 – unclear what is meant by 'this' here. Line 381 – unclear what is meant by 'it' here. | Accept - edited |
| Lines 393-394 – I understand what you are trying to say, but this is a very UK-centric take on diaries. I could point you to numerous personal diaries from beyond the UK that are biased towards drought. | Accept - We acknowledge that lines 392-394 presents a UK centric perspective and have provided examples from S.Africa and Mexico to demonstrate this bias towards extremes is not specific to this region or study |
| Lines 413-418 – I understand the reason for including this paragraph, but it has the potential to seriously detract from the results of the study. Much better would be to state earlier (in the methodology): (i) that one person completed the analysis for all index series to improve consistency; (ii) that volunteers were involved in transcription but there was quality control. | Agree- we have rephrased and inserted in section 2, thank you for the comments provided here. |
| Section 8 – I want to end on a positive. The results of this study represent a huge amount of work and are potentially very interesting. However, the Conclusion seriously underplays the quality of the research and should be much stronger. This is an opportunity to point out the key findings of the paper and highlight which aspects of the methodology were most successful. To my knowledge, no one has conducted this volume of analyses of diary entries, so this is the opportunity to recommend a methodology | Agree - We have redrafted the conclusions. We feel the revised version provides a stronger conclusion, and thank the referee for his constructive comments. |

| | |
|---|---|
| to be used in future studies. This will elevate the manuscript from 'an interesting study' to a 'must-cite paper' for future researchers seeking to use diaries for climate reconstruction. | |
| **Referee 2 comment** | **Author response** |
| [Evaluation] This manuscript presents very interesting research on reconstruction of rainfall amount from several simultaneous diaries in the past. The methods used are appropriate and the conclusions derived from these and the interpretations are consistent and sound. I believe the paper will be of interest to the readership of this journal and would recommend it for acceptance after the minor revisions. I look forward to seeing it in print. | Response to referee 2
We would like the thank the referee for reviewing the paper, and for their useful and constructive comments. We are glad that they consider the content of the paper to be sound. We are happy to incorporate their suggestions into our revised manuscript and have made an effort to include more relevant non-European literature. We have added more detail where suggested in the methodology and revised our figure captions. |
| [Comment] I understand that this paper deals with past climate in UK, Europe. However, in Chapter 1, I believe that related past studies should be introduced not only European cases but also other parts of the world (I could find only one reference in China). As far as I know, rain day index could show more strong correlation rather with temperature than rainfall amount in several Asian countries. I would like to recommend that the authors take more information from related studies outside of Europe and reflect it in the context. | Accept – we accept the comment from the referee and throughout the manuscript have attempted to link more and provide context to practices outside of Europe. |
| [Minor comment] Page 6, line 107-116: Please state more detailed classification and weather descriptions especially for Approach A and C. At this moment readers need to get and read references to understand the methods in detail. | Accept- Further examples and explanation have been added to this section. |
| Figure 1: In Figure caption, please explain what circles and triangles in the map refer. | Accept – clarity added to the caption |
| Figure 2: In Figure caption, please provide detail explanations. | Accept – caption expanded |

**Marked-up Manuscript:**

[revised manuscript text omitted]